# A Comprehensive Review of HER2 in Cancer Biology and Therapeutics

**DOI:** 10.3390/genes15070903

**Published:** 2024-07-11

**Authors:** Xiaoqing Cheng

**Affiliations:** Department of Oncology, School of Medicine, Washington University in Saint Louis, Saint Louis, MO 63108, USA; xiaoqing.cheng@wustl.edu

**Keywords:** HER2, cancer, metastasis, immune regulation, ADC, therapeutics

## Abstract

Human epidermal growth factor receptor 2 (HER2), a targetable transmembrane glycoprotein receptor of the epidermal growth factor receptor (EGFR) family, plays a crucial role in cell proliferation, survival, and differentiation. Aberrant HER2 signaling is implicated in various cancers, particularly in breast and gastric cancers, where HER2 overexpression or amplification correlates with aggressive tumor behavior and poor prognosis. HER2-activating mutations contribute to accelerated tumorigenesis and metastasis. This review provides an overview of HER2 biology, signaling pathways, mechanisms of dysregulation, and diagnostic approaches, as well as therapeutic strategies targeting HER2 in cancer. Understanding the intricate details of HER2 regulation is essential for developing effective targeted therapies and improving patient outcomes.

## 1. Introduction

### 1.1. HER2-Regulated Signaling Pathways and Their Role in Cellular Function

HER2 signaling pathways are critical regulatory systems in cellular function, particularly in cell growth, survival, and differentiation. Figure 1 summarizes the HER2 signaling pathways and their roles in cellular function.

The HER2 receptor has no known ligand that directly binds to it. Instead, HER2 is the preferred partner for forming heterodimers (pairing) with other EGFR family members [1]. HER2 dimerization is a crucial aspect of its signaling mechanism and plays a significant role in the biology of various cancers. The formation of dimers—either homodimers or heterodimers—is a key step in the activation of HER2 signaling pathways [2]. The nature and consequences of HER2 dimerization can vary significantly across different cancer types. In breast cancer, HER2 frequently forms heterodimers with HER3 [3]. The HER2/HER3 heterodimer is highly potent in activating downstream signaling pathways, such as PI3K/AKT and MAPK, promoting cell proliferation and survival. HER2 expression in gastric cancer can be more heterogeneous than in breast cancer, with variable patterns of expression within and between tumors. HER2 in gastric cancer also dimerizes with HER3 and potentially dimerizes with other EGFR family members, contributing to aggressive tumor behavior [4]. HER2 mutations, such as insertions in exon 20, can promote constitutive heterodimerization [5,6] and the activation of HER2 without ligand binding in non-small-cell lung cancer (NSCLC) [7]. HER2 can dimerize with EGFR in colorectal cancer, influencing responsiveness to EGFR-targeted therapies such as cetuximab [8]. HER2 is overexpressed in a subset of ovarian cancers, often forming dimers with other EGFR family members [9].

HER2 can form homodimers or heterodimers with other EGFR family members in a ligand-independent manner. HER2 homodimers are generally less potent in signaling than heterodimers. However, HER2/HER3 heterodimers are particularly potent, activating robust downstream signaling pathways [10]. HER3, with its six binding sites for the p85 subunit of PI3K, plays a critical role in PI3K/AKT pathway activation when dimerized with HER2 [11].

HER2 crosstalks with other signaling pathways through downstream targets. Akt activation not only promotes cell survival and growth but also crosstalks with other pathways. Akt can phosphorylate and activate mTOR (mammalian target of rapamycin), which regulates protein synthesis and cell growth [12]. Additionally, Akt can phosphorylate and inhibit TSC2 (tuberous sclerosis complex 2), activating the mTORC1 complex and the subsequent phosphorylation of downstream targets involved in protein synthesis [13]. HER2 activation can also lead to the activation of Ras, a small GTPase protein. Ras activates Raf, which initiates a cascade leading to the activation of ERK (extracellular signal-regulated kinase) [14]. ERK activation can crosstalk with the PI3K/Akt/mTOR pathway by phosphorylating and activating mTORC1 or directly phosphorylating and inhibiting TSC2, leading to mTORC1 activation [15]. ERK can also crosstalk with the JNK (c-Jun N-terminal kinase) pathway, influencing cellular responses such as apoptosis and proliferation [16,17]. HER2 signaling can crosstalk with the Wnt/β-catenin pathway, which is crucial in regulating tumor progression [18]. The activation of HER2 signaling can lead to the stabilization and nuclear translocation of β-catenin, a key mediator of Wnt signaling, promoting the transcription of Wnt target genes involved in cell proliferation and survival [19]. HER2 signaling can activate the NF-κB (nuclear factor kappa-light-chain-enhancer of activated B cells) pathway, which regulates inflammation, immunity, and cell survival genes [20]. The activation of HER2 signaling can lead to the phosphorylation and degradation of IκBα (inhibitor of NF-κB), resulting in the nuclear translocation of NF-κB and the transcriptional activation of its target genes [21]. In hormone receptor-positive breast cancer, HER2 signaling can crosstalk with estrogen receptor (ER) signaling, influencing the response to endocrine therapy [22]. HER2 activation can lead to the phosphorylation and activation of ER, enhancing its transcriptional activity and promoting cell proliferation [23]. 

Understanding the crosstalk between HER2 signaling and other pathways is critical for elucidating the complex mechanisms underlying cancer progression and treatment resistance. Targeting multiple signaling pathways simultaneously may be necessary to inhibit tumor growth and improve patient outcomes effectively.

### 1.2. Interaction with Other Members of the EGFR Family

HER2 (ErbB2) is unique because it can form homodimers (pairs) without ligand binding. This ligand-independent activity contributes to its role as an oncogenic driver in cancer. HER2 can form heterodimers with HER1 (EGFR). This interaction is significant, especially in cancer biology, as HER2/HER1 heterodimers have been shown to enhance oncogenic signaling pathways more potently than HER1 homodimers or HER2 homodimers alone [24]. 

HER2 has a unique relationship with HER3 (ErbB3). While HER3 has impaired kinase activity, it contains multiple docking sites for phosphotyrosine, making it an excellent partner for signaling amplification. HER2/HER3 heterodimers are particularly potent in activating downstream signaling pathways due to the strong recruitment and phosphorylation of HER3 by HER2 [25]. The HER2/HER3 heterodimer is especially important in prostate [26] and breast [27] cancers, where it enhances oncogenic signaling and contributes to the aggressive behavior of tumors [28].

HER2 can also form heterodimers with HER4 (ErbB4). Like HER3, HER4 lacks strong kinase activity but possesses multiple tyrosine phosphorylation sites. HER2/HER4 heterodimers have been implicated in mammary gland development, where they play a role in lactation [29]. However, their significance in cancer biology is less clear than HER2/HER1 and HER2/HER3 heterodimers. The ligand-independent activity of HER2 contributes to its role as an oncogenic driver in various cancers.

Understanding the intricate interactions between HER2 and other EGFR family members is crucial for elucidating the complexities of cellular signaling, particularly in cancer biology. Targeted therapies that disrupt these interactions, such as monoclonal antibodies and small-molecule inhibitors, have shown significant clinical efficacy in treating HER2-driven malignancies such as HER2-positive breast cancer.

## 2. Brief Overview of HER2 and Its Significance in Cancer Biology

HER2 is a receptor tyrosine kinase, a transmembrane protein of cells that helps regulate cell growth and division. HER2 can form homodimers or heterodimers with other EGFR family members in ligand-dependent and -independent manners. When HER2 and EGFR families form heterodimers to bind specific growth factors, signaling pathways that promote cell proliferation, survival, and migration are activated. In some cases, HER2 is overexpressed with too many copies of the HER2 gene, and, consequently, there is too much HER2 protein on the transmembrane of cancer cells. This overexpression can lead to uncontrolled cell growth and is associated with more aggressive forms of cancer [10]. 

Approximately 20–30% of breast cancers overexpress HER2. These tumors tend to be more aggressive and have a higher risk of recurrence than HER2-negative breast cancers. Some *HER2*-activating mutations were shown to be negative in the HER2 IHC clinical category with hyperactivated signaling [30], which can lead to metastatic breast cancer [31]. The discovery of HER2 mutations led to the development of targeted therapies designed to inhibit HER2 signaling. These therapies include drugs such as neratinib, trastuzumab (Herceptin), pertuzumab (Perjeta), ado-trastuzumab emtansine (Kadcyla), and trastuzumab deruxtecan (T-DXd). These drugs can effectively block HER2 signaling, slowing or halting the growth of HER2-positive breast cancer cells [32,33,34,35]. HER2 status is an essential prognostic and predictive marker in breast cancer. HER2-positive breast cancer tends to have a poorer prognosis than HER2-negative breast cancer [36]. However, targeted therapies have significantly improved outcomes for patients with HER2-positive breast cancer [37,38]. Adaptive and acquired drug resistance have been challenges in breast cancer [39,40,41], especially in HER2-mutated metastatic cancer [42,43,44]. Beyond breast cancer, HER2 alterations, including overexpression and activating mutations, have been identified in other cancer types, such as gastric and gastroesophageal cancers (4.4–53.4%) [45,46], non-small-cell lung cancer (4%) [47,48,49,50], endometrial cancer (20–40%) [51,52], and ovarian cancers [53]. Continued studies on HER2 and new therapies targeting HER2 signaling are being conducted in these cancers.

## 3. Dysregulation of HER2

### 3.1. Regulation of HER2 Expression and Activation and Mechanisms of Dysregulation

Like other receptors, the expression and activation of HER2 are tightly regulated by various transcriptional and post-translational mechanisms (Figure 2).

The expression of HER2 is primarily regulated at the level of gene transcription [54]. Transcription factors, such as Sp1 [55], AP-2 [56], and Ets-1 [57], can bind to specific regulatory elements in the HER2 gene promoter region or enhancer area, thereby activating or repressing HER2 gene transcription. Factors influencing transcription factor activity, such as growth factors, hormones, and oncogenes, can also indirectly affect HER2 expression levels. Epigenetic modifications, such as DNA methylation and histone acetylation, can regulate HER2 gene expression by modulating chromatin structure and accessibility to transcriptional machinery [58]. Aberrant epigenetic changes, such as the hypomethylation of the HER2 promoter region, can increase HER2 expression in cancer cells [59]. MicroRNAs (miRNAs) are small non-coding RNAs that can post-transcriptionally regulate gene expression by binding to the 3’ untranslated region (UTR) of target mRNAs and inhibiting translation or promoting mRNA degradation. Several miRNAs have been identified as regulators of HER2 expression, with some miRNAs suppressing HER2 expression in cancer cells [60].

Several regulatory proteins modulate HER2 activation by either promoting or inhibiting dimerization and/or tyrosine kinase activity. For example, the chaperone protein Hsp90 stabilizes HER2 and facilitates its maturation and activation [61]. Conversely, phosphatases such as PTPN12 can dephosphorylate HER2, attenuating its signaling activity [62]. The internalization of activated HER2 receptors via endocytosis serves as a mechanism for downregulating HER2 signaling [63]. Endocytosed receptors can undergo lysosomal degradation, leading to the attenuation of HER2-mediated signaling pathways.

The aberrant activation of signaling pathways, such as the Ras/Raf/MEK/ERK pathway or the PI3K/Akt pathway, can also upregulate HER2 transcription. Post-translational modifications, such as phosphorylation, ubiquitination, and glycosylation, can modulate HER2 protein stability, localization, and activity. The phosphorylation of HER2 tyrosine residues by kinases, such as Src or HER family members, can enhance HER2 signaling activity [64]. The ubiquitination of HER2 by E3 ubiquitin ligases can target HER2 for proteasomal degradation, regulating HER2 protein levels [65]. Aberrant interactions with other receptors or signaling molecules can enhance HER2 signaling activity and promote cancer progression [66]. 

Mutations in the HER2 gene can lead to alterations in the structure and function of the HER2 protein, contributing to cancer development and progression [30]. While HER2 gene amplification and overexpression are more common in cancer, mutations in HER2 have also been identified in certain cancer types. Mutations in the HER2 gene are relatively rare compared to HER2 gene amplification and overexpression. HER2 mutations have been identified in various cancer types, including breast cancer, gastric cancer, lung cancer, and colorectal cancer [67].

HER2 mutations can result in the constitutive activation of HER2 signaling pathways, leading to uncontrolled cell growth, proliferation, and survival [30]. Some HER2 mutations may confer resistance to HER2-targeted therapies [31]. Clinical trials are ongoing to evaluate the efficacy of HER2-targeted therapies in patients with HER2-mutant cancers, aiming to improve outcomes for these patients (DISNEY breast).

Overall, while less common than HER2 gene amplification, mutations in the HER2 gene can significantly impact cancer biology and treatment outcomes. Understanding the functional consequences of HER2 mutations and their clinical implications is essential for developing personalized treatment strategies for patients with HER2-mutant cancers. The dysregulation of HER2 expression and activation, often observed in cancer, underscores the importance of targeted therapies aiming to inhibit HER2 signaling in cancer treatment.

### 3.2. HER2 Crosstalk with Hormone Receptor Pathways

In breast cancer, HER2 signaling crosstalks with hormone receptor pathways, such as the estrogen receptor (ER) and the progesterone receptor (PR). This interaction is crucial in tumor development, progression, and therapeutic response. 

HER2 activation can enhance the transcriptional activity of hormone receptors, such as ER and PR, by activating downstream signaling pathways. HER2-mediated PI3K/Akt/mTOR pathway activation can phosphorylate and activate ER, leading to increased ER transcriptional activity and estrogen-dependent cell proliferation. HER2 signaling can also modulate the expression and function of co-regulatory proteins that interact with hormone receptors, further enhancing hormone receptor activity [23].

Hormone receptors, particularly ER, can also regulate the expression of HER2 in breast cancer cells. The estrogen stimulation of ER-positive breast cancer cells can induce HER2 expression, leading to increased HER2 signaling activity [68]. Conversely, the inhibition of ER signaling through hormone therapy (e.g., tamoxifen) or estrogen deprivation can downregulate HER2 expression and attenuate HER2 signaling [23]. Hormone receptor signaling can influence the response of HER2-positive breast cancer cells to HER2-targeted therapies, such as trastuzumab and lapatinib. The activation of ER signaling can promote resistance to HER2-targeted therapies by activating survival pathways and attenuating the cytotoxic effects of HER2 inhibition. Conversely, the inhibition of ER signaling, either alone or in combination with HER2-targeted therapies, can enhance the efficacy of HER2-targeted treatments by sensitizing HER2-positive breast cancer cells to therapy [22].

Combination therapies targeting both HER2 and hormone receptor pathways, such as the dual HER2 blockade (e.g., trastuzumab plus pertuzumab) and endocrine therapy (e.g., aromatase inhibitors or selective estrogen receptor modulators), have been shown to improve outcomes in patients with HER2-positive, hormone receptor-positive breast cancer. Biomarkers such as hormone receptor status and HER2 expression are used to guide treatment decisions and predict the response to targeted therapies in breast cancer patients. Understanding the complex crosstalk between HER2 and hormone receptor pathways is essential for developing optimal treatment strategies and improving outcomes for patients with HER2-positive breast cancer. Combination therapies targeting both HER2 and hormone receptor signaling pathways represent a promising approach for personalized treatment in this patient population.

## 4. HER2-Based Therapeutic Strategies and Future Directions

### 4.1. Evolution of HER2-Based Drugs 

The evolution of HER2-targeted drugs has significantly transformed the treatment landscape for HER2-positive cancers, particularly breast cancer. The development of these drugs has been a milestone in precision medicine, providing more effective and personalized treatment options. Figure 3 provides an overview of key HER2-targeted drugs and their evolution. Trastuzumab is a monoclonal antibody that binds to the extracellular domain of the HER2 protein, inhibiting its dimerization and signaling. It also induces antibody-dependent cellular cytotoxicity (ADCC). It was the first HER2-targeted therapy, and it significantly improved outcomes for HER2-positive breast cancer patients [69]. Lapatinib is a small tyrosine kinase inhibitor (TKI) that targets HER2 and EGFR (ErbB1). It inhibits the intracellular tyrosine kinase domain, blocking downstream signaling pathways. It is often used in combination with other therapies for patients with advanced or metastatic HER2-positive breast cancer who have progressed on trastuzumab [70]. Pertuzumab is a monoclonal antibody that binds to a different epitope on the HER2 extracellular domain from trastuzumab. It prevents the dimerization of HER2 with other ErbB receptors. It is typically used in combination with trastuzumab and chemotherapy, significantly enhancing antitumor activity [71]. T-DM1 is an antibody–drug conjugate that combines trastuzumab with a cytotoxic agent (DM1). It delivers the cytotoxic drug directly to HER2-positive cells, reducing systemic toxicity. It also provides the targeted delivery of chemotherapy, improving efficacy and reducing side effects [72]. Neratinib is an irreversible pan-HER TKI that inhibits HER1, HER2, and HER4. It is used as extended adjuvant therapy for early-stage HER2-positive breast cancer to reduce the risk of recurrence after the initial trastuzumab therapy [73]. Tucatinib is a highly selective HER2 TKI that blocks HER2 signaling pathways with a minimal impact on EGFR. It is often combined with trastuzumab and capecitabine to treat advanced unresectable or metastatic HER2-positive breast cancer [74]. Enhertu is an antibody–drug conjugate that combines trastuzumab with a topoisomerase I inhibitor payload. It provides a potent cytotoxic effect upon internalization by HER2-positive cells [75]. It has demonstrated significant efficacy in patients with advanced HER2-positive breast cancer, even those who previously received multiple lines of HER2-targeted therapies. Margetuximab is a monoclonal antibody similar to trastuzumab but engineered to enhance the immune system’s ability to kill cancer cells [76]. It is used in combination with chemotherapy for patients with advanced HER2-positive breast cancer who have received prior anti-HER2 therapies.

Bispecific antibodies are currently being developed to target HER2 and another antigen, potentially increasing efficacy simultaneously. Immunotherapy strategies are being explored to induce a more robust immune response against HER2-positive cancer cells. Research continues on next-generation TKIs and ADCs with improved efficacy and safety profiles. The continuous evolution of HER2-targeted therapies highlights the importance of understanding the molecular mechanisms of cancer and developing treatments that can more precisely target cancer cells while sparing normal tissues.

### 4.2. HER2 Diagnostic Approaches

Diagnosing HER2-positive breast cancer involves several approaches to assess HER2 expression or gene amplification accurately. Immunohistochemistry, fluorescence in situ hybridization, and chromogenic in situ hybridization are molecular techniques used to detect HER2 gene amplification in tumor cells. Like fluorescence in situ hybridization (FISH), chromogenic in situ hybridization (CISH) uses DNA probes that target the HER2 gene region, but the probes are labeled with chromogenic substances. Silver in situ hybridization (SISH) is a variation of CISH that uses silver staining to visualize HER2 gene amplification. SISH provides clear and crisp signals that can be easily interpreted by pathologists. Next-generation sequencing is a high-throughput molecular technique that can detect HER2 gene mutations and gene amplification and provide a comprehensive genomic profiling of tumors. NGS may be used in research settings or in clinical practice for patients with advanced or metastatic breast cancer to guide targeted therapy decisions. Multiplexed assays, such as the PAM50 gene expression assay or the Oncotype DX Breast Recurrence Score, may provide additional information about HER2 status and tumor biology, aiding in treatment decision-making and prognosis assessment.

These diagnostic approaches are crucial for accurately identifying HER2-positive breast cancer patients who may benefit from HER2-targeted therapies such as neratinib, trastuzumab deruxtecan, pertuzumab, and ado-trastuzumab emtansine (T-DM1). With advances in molecular profiling and biomarker testing, personalized treatment approaches based on HER2 expression, hormone receptor status, and genomic alterations are increasingly utilized to tailor therapy to individual patients. Biomarker-driven clinical trials evaluate targeted therapies and combination regimens in specific subgroups of HER2-positive breast cancer patients, such as those with HER2 mutations or amplifications, hormone receptor positivity, or specific genomic alterations.

Overall, HER2 status plays a critical role in guiding treatment decisions and predicting the response to therapy in breast cancer. Ongoing clinical trials continue to expand the treatment landscape for HER2-positive breast cancer, aiming to further improve the outcomes and quality of life of patients with this subtype of cancer. 

## 5. The Role of HER2 in Metastasis

Clinical studies have shown that HER2 overexpression, amplification, and mutations are associated with an increased metastatic potential and poor prognosis in breast, gastric, and ovarian cancers. HER2-positive breast cancer is associated with a higher risk of metastasis. 

HER2 activation leads to the activation of downstream signaling pathways, which promote cell cycle progression and proliferation. ERK is activated, which phosphorylates and activates transcription factors that promote cell cycle progression, such as c-Myc and cyclin D1. The HER2 activation of the PI3K/Akt/mTOR pathway leads to the phosphorylation and inactivation of pro-apoptotic proteins such as Bad [77], thereby preventing apoptosis. HER2 signaling can also activate the NF-κB pathway through AKT, which regulates cell survival and inflammation genes [20].

HER2 has been implicated in promoting metastasis. HER2 overexpression is associated with an increased invasiveness and metastatic potential in cancer cells. HER2 signaling promotes epithelial–mesenchymal transition (EMT) [78], a process in which epithelial cells acquire mesenchymal characteristics, allowing them to migrate and invade surrounding tissues. HER2 signaling can upregulate the expression of matrix metalloproteinases (MMPs) and facilitate cancer cell invasion and metastasis [79]. These enzymes degrade the extracellular matrix (ECM) and facilitate tumor cell invasion through the surrounding tissue, thus promoting invasion and intravasation. Once cancer cells invade through the ECM, HER2 signaling can also promote intravasation, the process by which cancer cells enter the blood or lymphatic vessels, enabling them to disseminate to distant organs.

Additionally, HER2 signaling can promote angiogenesis, the formation of new blood vessels, which is essential for supplying nutrients and oxygen to metastatic tumors, via the regulation of HIF1α [80]. Angiogenesis within the primary tumor microenvironment provides a blood supply to the growing tumor. Increased angiogenesis in the primary tumor creates a favorable microenvironment for cancer cell intravasation and dissemination into the bloodstream, promoting metastasis. Once cancer cells disseminate through the bloodstream or lymphatic system, HER2 signaling can promote their survival and colonization at distant metastatic sites. The HER2-driven PI3K/Akt/mTOR pathway enhances cancer cell survival and migration in the metastatic microenvironment, establishing secondary tumors [31].

Clinical studies have shown that HER2 is associated with an increased metastatic potential and poor prognosis in various cancers. Understanding the role of HER2 in metastasis is crucial for developing effective therapeutic strategies to target HER2-driven metastatic disease. HER2-targeted therapies have shown promise in treating HER2-positive metastatic cancers, highlighting the importance of HER2 as a therapeutic target in metastatic disease.

## 6. The Role of HER2 in Immune Regulation and Response

### 6.1. HER2 and Immune Response and Regulation

HER2-positive cancer has implications for the immune response within the tumor microenvironment [81]. HER2 is a tumor-associated antigen that can be recognized by the immune system [82]. The overexpression of HER2 on the transmembrane of cancer cells can lead to the presentation of HER2 peptides by major histocompatibility complex (MHC) molecules, eliciting an immune response [83]. HER2-targeted therapies, such as trastuzumab and pertuzumab, can enhance the immune recognition of HER2-positive tumor cells by promoting antibody-dependent cellular cytotoxicity (ADCC) and immune-mediated tumor cell death [84]. However, the details of the mechanism by which HER2 affects the ADCC and induces cell death are still unclear.

HER2-positive breast tumors often exhibit increased immune infiltration compared to HER2-negative tumors [85]. Tumor-infiltrating lymphocytes (TILs), particularly cytotoxic T cells, are commonly found in HER2-positive breast cancer [86,87]. The presence of TILs in HER2-positive breast tumors has been associated with an improved prognosis and response to HER2-targeted therapies, suggesting a role for the immune system in controlling HER2-positive breast cancer [88]. The expression of immune checkpoint molecules, such as programmed death-ligand 1 (PD-L1) and cytotoxic T-lymphocyte-associated protein 4 (CTLA-4), may modulate the immune response in HER2-positive breast cancer [89]. It is also unclear how the immune response is related to TILs and immune checkpoint molecules. Preclinical and clinical studies have shown that HER2-positive breast tumors can express PD-L1, which may contribute to immune evasion and resistance to HER2-targeted therapies [90]. Combination therapies targeting HER2 and immune checkpoints are being investigated in clinical trials to overcome immune evasion and enhance antitumor immune responses in HER2-positive breast cancer.

Immunotherapy, particularly immune checkpoint inhibitors targeting PD-1/PD-L1 or CTLA-4, has shown limited efficacy as monotherapy in HER2-positive breast cancer. However, approaches combining HER2-targeted therapies with immune checkpoint inhibitors or other immunomodulatory agents are being explored in clinical trials to enhance antitumor immune responses and improve outcomes for patients with HER2-positive breast cancer [90]. Currently, there is no combination of HER2 ADC therapy like T-DXd with other immune checkpoint inhibitors.

The JAK/STAT pathway, downstream of the HER2, is induced by the binding of NRG-1 and HER2/HER3 heterodimerization [91]. HER2-positive tumors often exhibit altered immune cell infiltration. HER2 signaling can attract immunosuppressive cells such as regulatory T cells (Tregs), myeloid-derived suppressor cells (MDSCs), and tumor-associated macrophages (TAMs), creating an immunosuppressive TME. HER2 signaling can upregulate cytokines such as IL-6 and chemokines such as CCL2 and CXCR4, promoting a pro-inflammatory and immunosuppressive environment conducive to tumor growth and metastasis [92]. HER2 signaling can induce the expression of PD-L1 on tumor cells, which bind to PD-1 on T cells, leading to T-cell exhaustion and immune evasion. HER2-positive tumors can downregulate MHC class I molecules, impairing antigen presentation and reducing the CTL-mediated killing of tumor cells [93]. NK cells can target HER2-positive tumors through antibody-dependent cellular cytotoxicity (ADCC). Therapeutic antibodies such as trastuzumab enhance ADCC by binding to HER2 on tumor cells and recruiting NK and macrophage cells to mediate tumor cell lysis [94]. HER2-targeted therapies can improve the presentation of HER2-derived peptides to T cells, boosting antitumor T-cell responses.

HER2 expression levels in breast cancer may serve as a potential biomarker for predicting the response to immune checkpoint inhibitors and other immunotherapies. Tumors with a high HER2 expression may be more immunogenic and responsive to immunotherapy approaches [95]. Addressing the immunosuppressive mechanisms associated with HER2 signaling can help overcome resistance to HER2-targeted therapies and immunotherapies. The rationale is that HER2-targeted therapies can enhance antigen presentation and T-cell activation, making tumors more susceptible to checkpoint blockade [96]. The interplay between HER2 signaling and immune regulation is complex and has significant implications for developing and optimizing cancer therapies. HER2-targeted treatments inhibit tumor growth directly and modulate the immune response, offering potential synergistic benefits when combined with immunotherapies. Ongoing research into this relationship is crucial for advancing therapeutic strategies and improving outcomes for patients with HER2-positive cancers.

Overall, HER2-positive breast cancer is associated with immune infiltration and interactions between HER2-targeted therapies and the immune system. Understanding the complex interplay between HER2 and the immune response is essential for developing effective immunotherapeutic strategies to improve outcomes for patients with HER2-positive breast cancer.

### 6.2. HER2 and TME

The tumor microenvironment (TME) plays a crucial role in the development, progression, and response to therapy in HER2-positive breast cancer. The TME of HER2-positive tumors is characterized by a complex interplay of various cell types, including cancer cells, stromal cells, immune cells, endothelial cells, and extracellular matrix components [97]. HER2-positive tumors often exhibit increased immune infiltration, with higher tumor-infiltrating lymphocyte (TIL) levels than HER2-negative tumors.

HER2-positive tumors can attract immune cells to the TME through various mechanisms, including the secretion of chemokines and cytokines. Tumor-infiltrating immune cells, such as T, B, natural killer (NK), and dendritic cells, interact with HER2-positive cancer cells and influence tumor growth, invasion, and metastasis [98]. Despite immune infiltration, HER2-positive tumors often exhibit immunosuppressive features within the TME, such as the expression of immune checkpoint molecules (e.g., PD-L1) and regulatory T cells (Tregs) [99]. These immunosuppressive mechanisms can impair antitumor immune responses and contribute to immune evasion and resistance to therapy in HER2-positive breast cancer. HER2 signaling can directly influence the composition and function of the TME in breast cancer. HER2 activation in cancer cells can produce cytokines, chemokines, and growth factors that modulate the TME [100]. HER2-positive breast tumors may exhibit alterations in angiogenesis, extracellular matrix remodeling, and immune cell recruitment due to dysregulated HER2 signaling.

Understanding the interactions between HER2 and the TME is essential for developing effective therapeutic strategies for HER2-positive breast cancer. Combination therapies targeting HER2 and TME components, such as immune checkpoint inhibitors, angiogenesis inhibitors, and stromal-targeting agents, are being investigated in preclinical and clinical studies to improve treatment outcomes in HER2-positive breast cancer [101]. The TME plays a critical role in shaping the behavior and response to therapy in HER2-positive breast cancer. Elucidating the complex interactions between HER2 signaling and TME components is essential for developing novel therapeutic approaches to overcome treatment resistance and improve outcomes for patients with HER2-positive breast cancer. Modulating HER2 signaling may impact immune regulation within the TME and enhance antitumor immune responses.

## 7. Conclusions and Future Perspectives

HER2 (human epidermal growth factor receptor 2) research continues to be a dynamic and rapidly evolving field, driven by the ongoing need to improve outcomes for patients with HER2-positive cancers. The development of next-generation ADCs aims to improve efficacy and reduce side effects compared to earlier ADCs such as T-DM1 and T-DXd. Bispecific antibodies are created to simultaneously target HER2 and another tumor-specific antigen, enhancing specificity and reducing resistance. Combining HER2-targeted therapies with immune checkpoint inhibitors (such as PD-1/PD-L1 inhibitors) aims to enhance antitumor immune responses, and combining HER2 inhibitors with drugs targeting other pathways (such as PI3K/AKT/mTOR) aims to overcome resistance and improve efficacy. The development of precision medicine and novel biomarkers for detection is needed. Genomic and proteomic profiling can be utilized to tailor HER2-targeted treatments to individual patients based on the specific molecular characteristics of their tumors. Identifying biomarkers that can predict which patients will benefit from HER2-targeted therapies or develop resistance will allow for more precise treatment adjustments.

Still, it is crucial to investigate the molecular mechanisms that lead to resistance to HER2-targeted therapies, such as activating mutations in the HER2 gene. Despite the substantial bulk, scRNA, and spatial transcriptomic data confirming their heterogeneity, there remains a lack of timing resolution regarding the phenotypic state during cancer and metastasis development and progression. This analytical gap prevents us from obtaining unbiased spatiotemporal information on dynamic changes in gene expression, limiting our comprehension of the functional roles played by HER2 during drug resistance and metastasis. Addressing this gap will significantly advance our knowledge and pave the way for developing more targeted therapeutical interventions.

The identification and targeting of new molecules involved in resistance mechanisms are required to develop novel therapies that bypass or overcome resistance. HER2-targeted therapies should be expanded to other cancers that express HER2, such as gastric, colorectal, and non-small-cell lung cancers. The use of nanoparticles and other advanced drug delivery systems improves the delivery and efficacy of HER2-targeted therapies, potentially reducing side effects and improving patient outcomes. The implementation of adaptive clinical trial designs that allow for modifications based on interim results enhances clinical trial efficiency and success rates. The leveraging of real-world data to complement clinical trials provides insights into the long-term effectiveness and safety of HER2-targeted therapies in diverse patient populations.

AI and machine learning can be utilized to develop predictive models for patient response to HER2-targeted therapies, aiding in the decision-making process for personalized treatment plans, and advanced computational methods can be employed to analyze large datasets from clinical trials and real-world evidence, uncovering new insights and potential therapeutic targets. The future of HER2 research lies in an integrated approach that combines advanced therapeutic strategies, precision medicine, and cutting-edge technologies. By addressing the challenges of resistance, expanding the indications for HER2-targeted therapies, and leveraging innovations in drug delivery and clinical trial design, researchers aim to significantly improve outcomes for patients with HER2-positive cancers.

## Figures and Tables

**Figure 1 genes-15-00903-f001:**
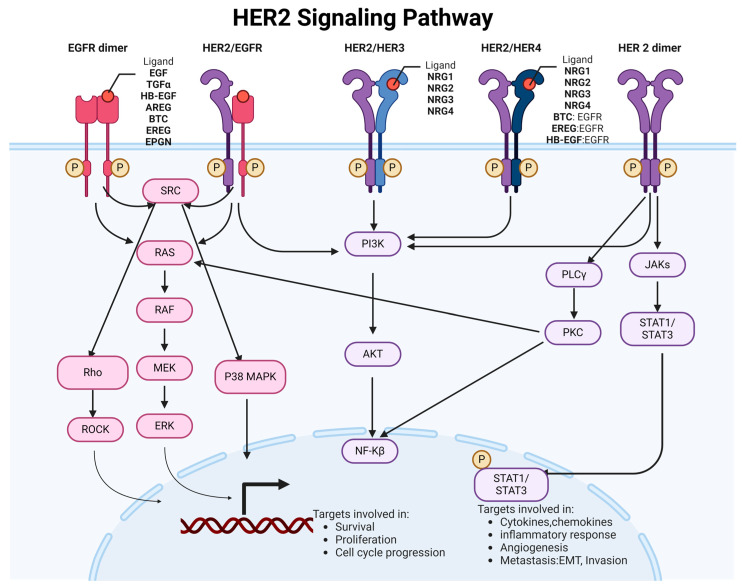
**Schematic representation of HER2 signaling regulation.** HER2 is the preferred partner for forming heterodimers (pairing) with other EGFR family members, such as HER1 (EGFR), HER3, and HER4. When a ligand binds to other EGFR family members, dimerization occurs, which activate the intrinsic tyrosine kinase activity of HER2, resulting in initiating downstream signaling cascades. These pathways contribute to regulating gene expression, cell cycle progression, cell motility, and other cellular functions. Figure was made using BioRender.

**Figure 2 genes-15-00903-f002:**
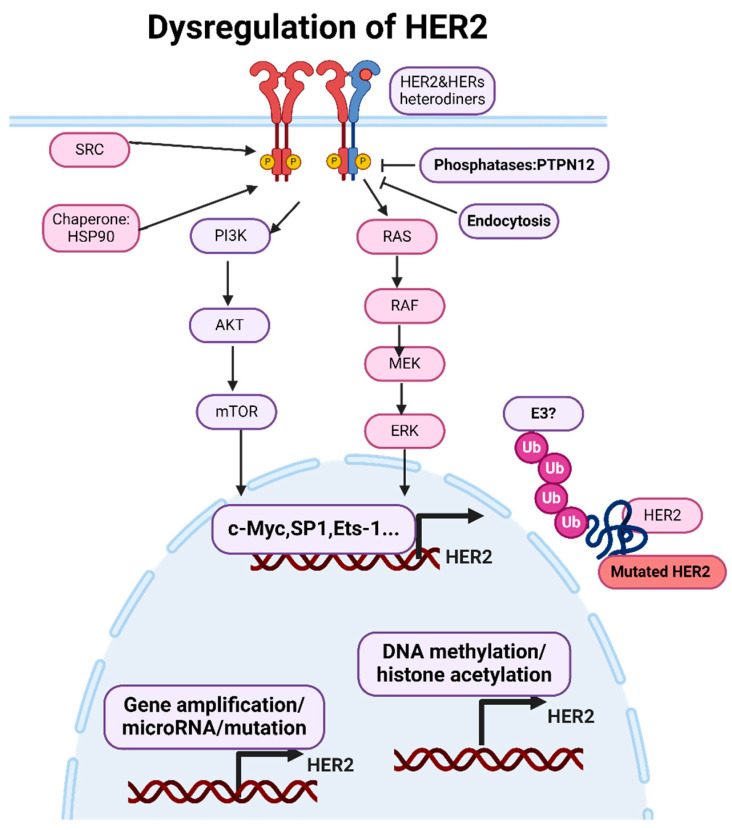
**Dysregulation of HER2. The expression of HER2 is primarily regulated at the level of gene transcription.** Transcription factors and epigenetic modifications can regulate HER2 gene expression by modulating the chromatin structure and accessibility to transcriptional machinery. Endocytosed receptors can undergo lysosomal degradation, leading to the attenuation of HER2-mediated signaling pathways. Figure was made using BioRender.

**Figure 3 genes-15-00903-f003:**
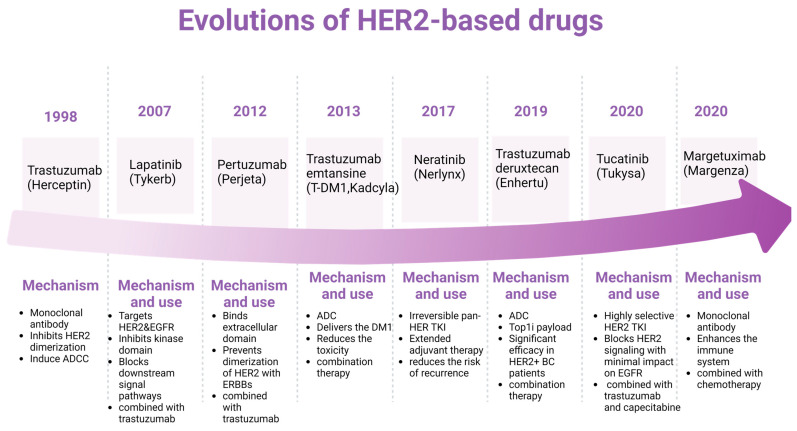
**Evolution of HER2-based drugs and mechanisms.** The timeline at the top shows when the corresponding HER2-based drugs were approved by the FDA. The corresponding mechanisms are at the bottom. Figure made using Biorender.

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
