# Peer review of "A Comprehensive Review of HER2 in Cancer Biology and Therapeutics"

_genes, 2024, doi:10.3390/genes15070903_

Round 1

Reviewer 1 Report

Comments and Suggestions for Authors

X. Chen reviews the HER2 receptor in cancer and different possibilities of successful therapeutics against breast cancer. This receptor is a member of the ErbB family of kinase receptors and is overexpressed in different types of cancer. Notably, it can be present in breast cancer, and it indicates a bad prognosis for patients. HER2 mutations can be targeted by different therapeutical molecules, as shown by the author. HER2 biology is an interesting topic; however, the review needs to be better organized for clarity and for the reader to follow all the information.

These are my comments:

HER2 is a receptor tyrosine kinase; it should say tyrosine kinase receptor or a receptor with tyrosine kinase activity.

The following statement: "When HER2 binds to specific growth factors, it activates signaling pathways that promote cell proliferation, survival, and migration." is not correct. HER2 does not bind to specific growth factors because it does not have a known ligand since it does not have an extracellular ligand binding site.

In some cases, HER2 is overexpressed with too many copies of the HER2 gene, and as a consequence, there is too much HER2 protein on the surface of cancer cells.

The section "HER2-regulated signaling pathways and its role in cellular function" should be the first section in the review to have a view and understand the function of the ErbB family receptors and to explain that HER2 does not have a known ligand and should form a heterodimer with other ErbB receptors, such as EGFR. This section should explain how HER2 homodimers can form and activate a signal to explain what is shown in  Figure 1. In this section, the full name of the receptor should be included, not in the conclusion.

In section 2.1, it is necessary to explain the interaction between HER2/HER3 and give more examples.

Section 2 should be: "Brief overview of HER2 and its significance in cancer biology"

In the paragraph, "The expression of HER2 is primarily regulated at the level of gene transcription. Transcription factors, such as Sp1, AP-2, and Ets-1, can bind to specific regulatory elements in the HER2 gene promoter region, cite, thereby activating or repressing HER2 gene transcription." Please explain the term "cite".

In subsection 6.1, the title needs to be completed. In this subsection, please describe further the combination of HER2-targeted therapies with immune checkpoint inhibitors.

In the following paragraph in subsection 6.1: "HER2-targeted therapies, such as trastuzumab and pertuzumab, can enhance immune recognition of HER2-positive tumor cells by promoting antibody-dependent cellular cytotoxicity (ADCC) and immune-mediated tumor cell death[65]." Please explain how antibodies, such as trastuzumab or pertuzumab, can promote ADCC or immune-mediated tumor cell death. How is this response related to TILs or immune checkpoint molecules?

In subsection  6.3, the statement "HER2 downstream, the JAK/STAT pathway, is Involved in immune response regulation and can influence inflammation." The author mentions that this pathway is activated downstream of HER2; however, the relationship of the JAK/STAT pathway with all the molecules described in this subsection is not clear.

The information described in the subsections of section 6 is redundant. It should be better organized.

The figures should be organized and ordered according to the text description.

Author Response

Dear reviewer,

Thank you for your comments. The following is my list of revisions. Please see the revised version to match your corresponding comments one by one. Thank you!

Comments:

HER2 is a receptor tyrosine kinase; it should say tyrosine kinase receptor or a receptor with tyrosine kinase activity.

Yes, this is fixed in the revised manuscript. I have used the term “tyrosine kinase receptor”.

The following statement: "When HER2 binds to specific growth factors, it activates signaling pathways that promote cell proliferation, survival, and migration." is not correct. HER2 does not bind to specific growth factors because it does not have a known ligand since it does not have an extracellular ligand binding site.

I have made modifications to this statement. Please see the revised one in our manuscript. “HER2 can form homodimers or heterodimers with other EGFR family members in a ligand-dependent and independent manner. When HER2 and EGFR families form heterodimers to bind specific growth factors, signaling pathways that promote cell proliferation, survival, and migration are activated.”

In some cases, HER2 is overexpressed with too many copies of the HER2 gene, and as a consequence, there is too much HER2 protein on the surface of cancer cells.

I have modified this statement into “In some cases, HER2 is overexpressed with too many copies of the HER2 gene, and consequently, there is too much HER2 protein on the surface of cancer cells”. Please see the revised one in our manuscript.

The section "HER2-regulated signaling pathways and its role in cellular function" should be the first section in the review to have a view and understand the function of the ErbB family receptors and to explain that HER2 does not have a known ligand and should form a heterodimer with other ErbB receptors, such as EGFR. This section should explain how HER2 homodimers can form and activate a signal to explain what is shown in  Figure 1. In this section, the full name of the receptor should be included, not in the conclusion.

In section 2.1, it is necessary to explain the interaction between HER2/HER3 and give more examples.

Section 2 should be: "Brief overview of HER2 and its significance in cancer biology"

Yes. Section 2 and Section 1 are reversed. Also, in reponse to your suggestions about HER2/HER3 interaction, I revised the statements as “HER2/HER3 heterodimers are particularly potent in activating downstream signaling pathways due to the strong recruitment and phosphorylation of HER3 by HER2[25]. The HER2/HER3 heterodimer is especially important in prostate[26] and breast[27] cancers to enhance oncogenic signaling and contribute to the aggressive behavior of tumors.”

In the paragraph, "The expression of HER2 is primarily regulated at the level of gene transcription. Transcription factors, such as Sp1, AP-2, and Ets-1, can bind to specific regulatory elements in the HER2 gene promoter region, cite, thereby activating or repressing HER2 gene transcription." Please explain the term "cite".

The statement has been revised as “The expression of HER2 is primarily regulated at the level of gene transcription. Transcription factors, such as Sp1, AP-2, and Ets-1, can bind to specific regulatory elements in the HER2 gene promoter region or enhancer area, thereby activating or repressing HER2 gene transcription.”

In subsection 6.1, the title needs to be completed. In this subsection, please describe further the combination of HER2-targeted therapies with immune checkpoint inhibitors.

Yes. The title has already been completed.

In the following paragraph in subsection 6.1: "HER2-targeted therapies, such as trastuzumab and pertuzumab, can enhance immune recognition of HER2-positive tumor cells by promoting antibody-dependent cellular cytotoxicity (ADCC) and immune-mediated tumor cell death[65]." Please explain how antibodies, such as trastuzumab or pertuzumab, can promote ADCC or immune-mediated tumor cell death. How is this response related to TILs or immune checkpoint molecules?

Thank you for pointing out these two questions. However, how antibodies, such as trastuzumab or pertuzumab, can promote ADCC or immune-mediated tumor cell death and how this response is related to TILs or immune checkpoint molecules is still unclear. To clarify this, I added the statement, “However, the details of the mechanism of how HER2 affects the ADCC and induces cell death are still unclear.”

In subsection  6.3, the statement "HER2 downstream, the JAK/STAT pathway, is Involved in immune response regulation and can influence inflammation." The author mentions that this pathway is activated downstream of HER2; however, the relationship of the JAK/STAT pathway with all the molecules described in this subsection is not clear.

Thank you for mentioning this. To specified the description, I edited the statement into “HER2 downstream, the JAK/STAT pathway, is induced by the binding of NRG-1 and HER2/HER3 heterodimerization [71].”

The information described in the subsections of section 6 is redundant. It should be better organized.

Yes, I combined the subparagraph that is redundant in section 6 and reorganized it.

The figures should be organized and ordered according to the text description.

Yes, the figures are reorganized according to the text description.

Thank you for all your constructive comments!

Reviewer 2 Report

Comments and Suggestions for Authors

I have thoroughly reviewed your manuscript titled " A Comprehensive Review of HER2 in Cancer Biology and Therapeutics " I would like to commend the author on a well-structured and insightful review. This manuscript provides valuable contributions to the understanding of the biology of HER2 in its possible role in the therapy. It is a big summary of a lots if references.

The only thing I would suggest:

Please shorten the Figures legend or transfer some portions into the main text, it could mislead the reader.

HER2 was described by author in many caner types. However, there is no information its expression and role in the pathology of renal cancer. Is there any valuable information about it in the literature, if  so, please add in a short section

Author Response

Thank you for your comments.

Your questions about HER2 in renal cancer is great. Renal cancer is one of my interests and will be one of my future directions. Among all the oncogenes across the renal cancer subtypes, HER2 is not on the top genes list. Also, HER2 is usually expressed in normal renal tissue but not renal cell carcinoma tissue. That's the main reason I didn't mention these cancer type in this review.

For your suggestion about figure legend,we have shorten it and transfer some part to the main text base on your suggestion. Please see the changes in the revised manuscript.

Thanks again for your advice!

Reviewer 3 Report

Comments and Suggestions for Authors

Here, the authors presented the role HER2, a transmembrane protein receptor, has in cancer progression, prognosis, therapy approaches, and diagnostics. Similar papers have been published; however, this is the authors' point of view.

Important concerns to address are as follows:

I suggest “transmembrane protein” instead of “surface protein”

I suggest clarifying “There are 5% HER2 activating mutations, which were shown negatively in the HER2 IHC clinical category with hyperactivated signaling.[2], which can lead to metastatic breast cancer[3].”.

Instead of a long figure legend for figures 1 and 2, I strongly suggest adding the specified mechanisms in the figure itself. The schematic representation is much easier to follow than giving part of the information in the schematic and part in the long text below the figure.

Figures are an integral part of the manuscript and facilitate the reading of the manuscript. In a review paper, the schematic representation of molecular mechanisms is necessary, so that the explanation of the mechanism, especially in the Introduction, does not draw the reader's attention away from the manuscript's key subject. The figures need to be integrated and mentioned in the text appropriately.

“HER2 can dimerize with EGFR in colorectal cancer, influencing responsiveness to EGFR-targeted therapies like cetuximab” – please add how the HER2-EGFR dimer in colorectal cancer influences the responsiveness to cetuximab (or other EGFR-targeted therapies).

Please specify what exactly you mean by “potency” in terms of HER2 homo- or heterodimers.

The whole “crosstalk” among signaling pathways section should be represented graphically (4th paragraph in section 2. HER2-regulated signaling pathways and their role in cellular function).

In section 2, the text appears randomly subdivided into subsections. E.g. you write about HER2/HER3 dimer; then, on the interacting signaling pathways of importance in cancer; then return to HER2/HER1 dimer in the subsection “Interaction with other members of the EGFR family”; and then come back again to HER2/HER3. I strongly suggest reorganizing the text structure in section 2, to avoid repetitive statements and jumping at random from one subject to the next.

References to the research supporting the statements are missing altogether in section 3. In section 6.2 references are missing as well.

Figure 4 is mentioned in the text in the introduction, in the second paragraph. And yet, it is mentioned in the text on page 6, and before Figure 3 – the current Figure 4 should be placed near the text where it is mentioned first. The figures should be numbered in order of appearance and mentioned in the text where appropriate.

Current section 5 should be after section 3. Then, proceed to diagnostics and therapy (current section 4). This is another example of jumping from one subject to the next, with a poor connection between the sections.

Please check the heading of section 6.1, it appears incomplete.

In “Conclusions and Future Perspectives”, starting with paragraph 3 it appears the authors are listing the perspectives when the text should be in narrative. 

Comments on the Quality of English Language

The text needs to have extensive English language proofing.

Among other issues with the English language, in several sections, sentence structure is in listing form, when it should be in narrative writing. There are misplaced full stops along the text, especially before the citations. Also, spaces are missing before citations. There is a misuse of capital letters as well.

Author Response

Response:

Dear Review,

Thank you for your comments. I appreciate your suggestions. One big revision corresponding to your comments is the English editing. Please see the revised version. In addition, I also revised the manuscript based on most of your suggestions one by one. 

Important concerns to address are as follows:

I suggest “transmembrane protein” instead of “surface protein”

Yes. I have changed all the “surface protein” into “transmembrane protein” in our revised manuscript.

I suggest clarifying “There are 5% HER2 activating mutations, which were shown negatively in the HER2 IHC clinical category with hyperactivated signaling.[2], which can lead to metastatic breast cancer[3].”

Thank you for pointing out this. Please see the revised statement: “Several HER2 activating mutations were shown negatively in the HER2 IHC clinical category with hyperactivated signaling[29], which can lead to metastatic breast cancer[30].”

Instead of a long figure legend for figures 1 and 2, I strongly suggest adding the specified mechanisms in the figure itself. The schematic representation is much easier to follow than giving part of the information in the schematic and part in the long text below the figure.

Yes. I have shortened the figure legend length for figures 1 and 2.

Figures are an integral part of the manuscript and facilitate the reading of the manuscript. In a review paper, the schematic representation of molecular mechanisms is necessary, so that the explanation of the mechanism, especially in the Introduction, does not draw the reader's attention away from the manuscript's key subject. The figures need to be integrated and mentioned in the text appropriately.

Yes. I have reorganized the figures and text properly.

“HER2 can dimerize with EGFR in colorectal cancer, influencing responsiveness to EGFR-targeted therapies like cetuximab” – please add how the HER2-EGFR dimer in colorectal cancer influences the responsiveness to cetuximab (or other EGFR-targeted therapies).

Yes. Please see the revised statement: “HER2 can dimerize with EGFR in colorectal cancer, causing resistance to EGFR-targeted therapies like cetuximab by activating the downstream signaling pathway”.

Please specify what exactly you mean by “potency” in terms of HER2 homo- or heterodimers.

Good question. Yes. Please see the related statements: “HER2 homodimers are generally less potent in signaling than heterodimers. However, HER2-HER3 heterodimers are particularly potent, activating robust downstream signaling pathways[10]. HER3, with its six binding sites for the p85 subunit of PI3K, plays a critical role in PI3K/AKT pathway activation when dimerized with HER2[11].”

The whole “crosstalk” among signaling pathways section should be represented graphically (4th paragraph in section 2. HER2-regulated signaling pathways and their role in cellular function).

Thank you for your kind advice. I didn’t graph this because some of these crosstalk pathways overlapped with Figure 1.

In section 2, the text appears randomly subdivided into subsections. E.g. you write about HER2/HER3 dimer; then, on the interacting signaling pathways of importance in cancer; then return to HER2/HER1 dimer in the subsection “Interaction with other members of the EGFR family”; and then come back again to HER2/HER3. I strongly suggest reorganizing the text structure in section 2, to avoid repetitive statements and jumping at random from one subject to the next.

Yes. Revised base on your suggestion. Please see the detail in the revised manuscript.

References to the research supporting the statements are missing altogether in section 3. In section 6.2 references are missing as well.

Yes. References are added to the section 3 and 6.2.

Figure 4 is mentioned in the text in the introduction, in the second paragraph. And yet, it is mentioned in the text on page 6, and before Figure 3 – the current Figure 4 should be placed near the text where it is mentioned first. The figures should be numbered in order of appearance and mentioned in the text where appropriate.

Yes. The figures are reorganized correspondingly.

Current section 5 should be after section 3. Then, proceed to diagnostics and therapy (current section 4). This is another example of jumping from one subject to the next, with a poor connection between the sections.

Good point. I understand your meaning. I keep this because the therapy for HER2-positive metastatic cancers still needs to be improved. Currently, the diagnostics and therapy in section 4 are still not fully conclusive in HER2-positive cancer metastasis.

Please check the heading of section 6.1, it appears incomplete.

Yes. Completed.

 In “Conclusions and Future Perspectives”, starting with paragraph 3 it appears the authors are listing the perspectives when the text should be in narrative.

I have revised this paragraph to improve it.”The identification and targeting of new molecules involved in resistance mechanisms are required to develop novel therapies that bypass or overcome resistance. HER2-targeted therapies should be expanded to other cancers that express HER2, such as gastric, colorectal, and non-small-cell lung cancers. The use of nanoparticles and other advanced drug delivery systems improves the delivery and efficacy of HER2-targeted therapies, potentially reducing side effects and improving patient outcomes. The implementation of adaptive clinical trial designs that allow for modifications based on interim results enhances clinical trial efficiency and success rates. The leveraging of real-world data to complement clinical trials provides insights into the long-term effectiveness and safety of HER2-targeted therapies in diverse patient populations.”

Round 2

Reviewer 1 Report

Comments and Suggestions for Authors

The paper is more organized and the reader will find it useful.

Please indicate the full names for FISH and CISH.

In the 6.1 subsection, the JAK-STAT pathway is downstream of HER2.

HER2 is a receptor; therefore it is upstream of the JAK-STAT pathway. Please correct the sentence.

Comments on the Quality of English Language

English language is fine, minor editing of English requiered. For example: Evolutions of HER-2 based drugs;

Combination therapies targeting....

Author Response

Dear reviewer,

Thank you for your comments.

  1. The full name for FISH and CISH is added.
  2. Please see the revised version: ‘‘JAK/STAT pathway, downstream of the HER2, is induced by the binding of NRG-1 and HER2/HER3 heterodimerization [74].”

Thank you!

Reviewer 3 Report

Comments and Suggestions for Authors

The quality of English language is significantly improved. The structure of the manuscript is improved. The manuscript is comprehensive and understandable.  The author addressed some of the previously made comments. References are still missing in the sections previously mentioned. 

Author Response

Dear reviewer,

Thank you for your comments.

More references are added to section 3 and section 6.2.